# Geographic Monitoring of Insecticide Resistance Mutations in Native and Invasive Populations of the Fall Armyworm

**DOI:** 10.3390/insects12050468

**Published:** 2021-05-18

**Authors:** Sudeeptha Yainna, Nicolas Nègre, Pierre J. Silvie, Thierry Brévault, Wee Tek Tay, Karl Gordon, Emmanuelle dAlençon, Thomas Walsh, Kiwoong Nam

**Affiliations:** 1DGIMI, University of Montpellier, INRAE, F-34095 Montpellier, France; sudeepha.yainna-kumarihami@inrae.fr (S.Y.); nicolas.negre@umontpellier.fr (N.N.); emmanuelle.d-alencon@inrae.fr (E.d.); 2CIRAD, UPR AIDA, F-34398 Montpellier, France; pierre.silvie@cirad.fr; 3PHIM Plant Health Institute, University of Montpellier, IRD, CIRAD, INRAE, Institut Agro, F-34398 Montpellier, France; thierry.brevault@cirad.fr; 4AIDA, University of Montpellier, CIRAD, F-34398 Montpellier, France; 5Black Mountain Laboratories, CSIRO, Canberra, ACT 2601, Australia; weetek.tay@csiro.au (W.T.T.); karl.gordon@csiro.au (K.G.)

**Keywords:** ABCC2, *Bacillus thuringiensis*, biological invasion, Cytochrome P450, Fall armyworm, insecticide resistance, *Spodoptera frugiperda*

## Abstract

**Simple Summary:**

The moth fall armyworm (*Spodoptera frugiperda*) is a major agricultural pest insect damaging a wide range of crops, especially corn. Field evolved resistance against *Bacillus thuringiensis* (Bt) toxins and synthetic insecticides has been repeatedly reported. While the fall armyworm is native to the Americas, its biological invasion was first reported from West Africa in 2016. Since then, this pest has been detected across sub-Saharan and North Africa, Asia, and Oceania. Here, we examine the geographical distribution of mutations causing resistance against Bt or synthetic insecticides to test if the invasion was accompanied by the spread of resistance mutations using 177 individuals collected from 12 geographic populations including North and South America, West and East Africa, India, and China. We observed that Bt resistance mutations generated in Puerto Rico or Brazil were found only from their native populations, while invasive populations had higher copy numbers of cytochrome P450 genes and higher proportions of resistance mutations at AChE, which are known to cause resistance against synthetic insecticides. This result explains the susceptibility to Bt insecticides and the resistance against synthetic insecticides in invasive Chinese populations. This information will be helpful in investigating the cause and consequence associated with insecticide resistance.

**Abstract:**

Field evolved resistance to insecticides is one of the main challenges in pest control. The fall armyworm (FAW) is a lepidopteran pest species causing severe crop losses, especially corn. While native to the Americas, the presence of FAW was confirmed in West Africa in 2016. Since then, the FAW has been detected in over 70 countries covering sub-Saharan Africa, the Middle East, North Africa, South Asia, Southeast Asia, and Oceania. In this study, we tested whether this invasion was accompanied by the spread of resistance mutations from native to invasive areas. We observed that mutations causing Bt resistance at ABCC2 genes were observed only in native populations where the mutations were initially reported. Invasive populations were found to have higher gene numbers of cytochrome P450 genes than native populations and a higher proportion of multiple resistance mutations at acetylcholinesterase genes, supporting strong selective pressure for resistance against synthetic insecticides. This result explains the susceptibility to Bt insecticides and resistance to various synthetic insecticides in Chinese populations. These results highlight the necessity of regular and standardized monitoring of insecticide resistance in invasive populations using both genomic approaches and bioassay experiments.

## 1. Introduction

Insecticide resistance is one of the main challenges for the control of insect pests. Commonly used insecticides to control pest insects can be classified into two main types. The first group is Bt (*Bacillus thuringiensis*) toxins, which are generally produced by transgenic crops. The global areas of planted Bt crops is positively correlated with the number of Bt resistance pest species, implying that field-evolved Bt resistance is prevalent [1]. Field-evolved Bt resistance is a particularly serious issue in corn and cotton because the majority of those plants are Bt crops in the USA and because only GM-cotton expressing Bt toxins are grown in Australia. Field-evolved Bt resistance has been observed in several major insect Lepidoptera pest species including maize stalk borer (*Busseola fusca*), western corn rootworm (*Diabrotica virgifera virgifera*), cotton bollworm (*Helicoverpa zea*), Old World cotton bollworm (*Helicoverpa armigera*), *Helicoverpa punctigera*, fall armyworm (*Spodoptera frugiperda*), and pink bollworm (*Pectinophora gossypiella*) [1]. The other group is synthetic insecticides such as organophosphates, pyrethroids, neonicotinoids, organochlorides, carbamates, ryanoids, and spinosyns. By 2021, more than 600 arthropod species have been included in the arthropod pesticide resistance database (https://www.pesticideresistance.org/) (accessed on 21 January 2021).

Several genetic mechanisms of Bt resistance have been reported (reviewed in [2]). If a mutation in a midgut receptor causes reduced physical binding with Bt proteins, the toxicity of Bt can be decreased. These receptors include cadherin, aminopeptidase-N, alkaline phosphatases, and ATP-binding cassette transporters. Alternatively, altered processing of Bt prototoxin, increased immune status, sequestration of Bt protein, and accelerated recovery of epithelial integrity can also increase Bt resistance. The resistance against synthetic insecticides can be caused by detoxification genes [3] such as cytochrome P450 gene, esterase, and glutathione S transferase. In addition, mutations in calcium channels, acetylcholinesterase, and nicotinic acetylcholine receptors may also cause resistance [4,5,6,7].

Fall armyworm (FAW, *Spodoptera frugiperda,* J.E.Smith) (Lepidoptera: Noctuidae: Noctuinae) is one of the most damaging pest insects, partly due to the extreme polyphagy by feeding on at least 353 species in 76 plant families, which include several economically important crops such as corn, rice, sorghum, sugarcane, cotton, and soybean [8] Genomic analyses have demonstrated a rapid expansion of detoxification genes [9,10,11], potentially associated with this extreme polyphagy by overcoming plant defense toxins from diverse plants.

Field-evolved resistance to Bt insecticide in the FAW has been reported from Puerto Rico [12,13], Brazil [14,15], Argentina [16], and the USA [17]. Five mutations causing Bt resistance have been reported. Interestingly, all these mutations were observed in the ATP Binding Cassette Subfamily C Member 2 (ABCC2) gene, which encodes ATP-binding cassette transporter proteins, implying parallel evolution. In a population from Puerto Rico, 2bp GC insertion at ABCC2 causes a frameshift mutation and a premature stop codon, which results in improper binding between ABCC2 and Bt Cry1F and Cry1A proteins [18,19]. Boaventura et al. identified three causal mutations of Cry1F resistance at ABCC2 including GY deletion, P799K/R substitution, and G1088D substitution from a Brazilian population [20]. Guan et al. also identified a 12bp insertion, which causes a frameshift mutation and a premature stop codon, from another Brazilian population [21]. As well as the Bt resistance, field-evolved resistance to synthetic insecticides has been widely reported. For example, a population in Puerto Rico was reported to have resistance against a wide range of synthetic insecticides including flubendiamide, chlorantraniliprole, methomyl, thiodicarb, permethrin, chlorpyriphos, zeta-cypermethrin, deltamethrin, triflumuron, and spinetoram [22]. We observed that P450 genes cause resistance against deltamethrin [23]. Boaventura et al. reported that I4734M and G4946E substitutions at the ryanodine receptor (RyR) cause resistance against diamides [24]. Carvalho et al. reported that resistance against organophosphate and carbamates is caused by A314S, G340A, and F402V mutations at acetylcholinesterase (AChE) and that pyrethroid resistance is caused by T929I, L932F, and L1014F mutations at the voltage-gated sodium channel (VGSC) gene [4].

FAW is native to North and South America, and its presence was first confirmed in West Africa in 2016 [25]. Since then, FAW has been detected across almost the entire Sub-Saharan Africa, in northern Africa/the Near East (Egypt), Middle East Asia, South Asia, South East Asia, East Asia, Australia, and more recently, the Canary Islands located off the coast between Western Africa and Morocco, and New Caledonia in the South Pacific (https://www.cabi.org/ISC/fallarmyworm) (accessed on 21 January 2021). The invasion caused a serious reduction in corn production, between 21–53% in Africa [26]. While native populations are composed of two sympatric strains, corn (sfC) and rice strains (sfR), named after their preferred host-plants [27,28,29], the invasive populations are observed as hybrids between corn and rice strains [30,31,32], but are predominantly observed in corn fields. Genomic analyses support the multiple introductions of FAW from multiple native populations including Florida and Brazil to multiple invasive areas including China and Africa [31,33]. However, the invasive populations exhibited much more homogeneous DNA sequences than native populations, implying that these invasive populations had experienced extensive admixture since their establishment across the Old World [31].

The invasion process opens up the possibility that resistance mutations have been spread as well in the invasive area. Recently, Boaventura et al. observed that resistance mutations at AChE were frequently observed from two invasive populations in Kenya and Indonesia and two populations in Brazil and Puerto Rico, while GY deletion of ABCC2 genes was observed only in the Brazilian population [34]. Boaventura et al. also reported the presence of the VGSC L1014F allele with low frequency in an Indonesian population that has, to-date, yet to be reported in other invasive populations [34]. Zhang et al. reported that the 2 bp GC insertion was not observed from Chinese populations [32], while Guan et al. reported that the 12 bp insertion in exon 15 of the ABCC2 gene was not detected in FAW populations from Uganda, Malawi, and from six Chinese provinces [21]. However, we still lack a comprehensive understanding of the geographic distribution of these resistance mutations because global profiling of the causal resistance mutations has not yet been performed. We performed whole-genome resequencing from 177 individuals from both invasive and native populations sampled from 12 geographic populations in North America, South America, Africa, India, and China to investigate adaptive evolution associated with the invasive success in our previous study [31]. In this study, we undertook reanalysis of the resequencing data of the FAW populations to test the presence of causal resistance mutations against insecticides in the FAWs.

## 2. Methods

### 2.1. Resequencing Data

Resequencing data generated in our previous study were reused [31,33]. More specifically, this dataset included 29 sfC and 49 sfR individuals from invasive populations (Benin, Malawi, Uganda, China, and India) and 70 sfC and 29 sfR individuals from native populations (Mississippi, Florida, Puerto Rico, Brazil, French Guinea, Guadeloupe, and Mexico) (Figure 1) (please see Appendix A for more information). This dataset includes four Bt-resistant and six susceptible Brazilian individuals. Paired-end whole genome resequencing was performed with approximately 20X coverage with 300 bp insert size, and 150 bp read length using Illumina (San Diego, CA, USA) HiSeq 2500, HiSeq 4000, and Novaseq 6000. Variant calling was performed using the GATK-4.0.11.0 package [35] for single nucleotide variant (SNV) and CNVCaller [36] for copy number variation (CNV). The numbers of variations were 27,117,672 and 22,916 for SNV and CNV, respectively.

### 2.2. Gene Annotation and Statistical Analysis

ABCC2, RyR, AChE, and VGSC were annotated by aligning protein sequences obtained from NCBI and reference genomes using exonerate 2.2.0 [38] with the protein2genome model. The accession numbers of these genes were QGS83596, XP_022819835, MK226188, and KC435025, for ABCC2, AChE, RyR, and VGSC, respectively. Genetic variations within these genes were identified from the resequencing data [31] using IGV tools [39]. P450 genes were identified from the gene annotation files generated in our previous study (OGS7.0) [31]. The identification of the P450 clan was performed by blasting against OGS2.2 [9], which includes manually annotated P450 genes, with 98% cutoff using ncbi-blast-2.10.0+ [40]. The Fisher’s exact test was performed using the R package to test a statistical significance of increased gene duplication of P450 genes. Weir and Cockerham’s F_ST_ [41] between Bt susceptible and resistant individuals was calculated at the ABCC2 gene using vcftools v0.1.13 [42]. Increased F_ST_ at ABCC2 was tested by calculating the proportion of F_ST_ at randomly chosen loci with the same length of ABCC2 from whole genome sequences with 100,000 replications.

## 3. Results

### 3.1. Bt Insecticide Resistance–ABCC2 Gene

We tested the presence of five reported mutations at the ABCC2 gene causing the resistance against Bt insecticide. These mutations include (i) 2 bp GC insertion leading to a premature stop codon due to the frameshift (originally identified from a population in Puerto Rico) [18,19]; (ii) GY deletion (Brazil) [20]; (iii) 12 bp insertion causing premature stop codon due to frameshift (Brazil) [21]; (iv) P799K/R substitution (Brazil) [20]; and (v) G1088D substitution (Brazil) [20] (Figure 2). We identified the 2 bp GC insertion from 11 individuals from Puerto Rico in the resequencing data (Table 1). As this mutation was originally reported from a population in Puerto Rico, this result supports that this mutation remained in this original population in Puerto Rico and did not spread to the Old World by invasion.

GY deletion was identified from two individuals in the Brazilian population, and P799K/R was observed from the same individuals. This result implies that this mutation also remained in the original population. Intriguingly, these mutations exist at the homozygous state in these two Bt susceptible individuals, suggesting a possibility that these mutations alone are not sufficient for Bt resistance.

We identified two individuals with 12 bp insertion only from Brazil, also supporting that this mutation remains in the original population and confirms the analysis of Guan et al. [21]. G1088D was not observed from the resequencing data.

As unidentified mutations causing Bt resistance might exist, we investigated sequences with complete genetic differentiation between resistant and susceptible Brazilian individuals. In total, 49 SNPs had complete genetic differentiation (i.e., F_ST_ = 1), and all these positions were found to be intronic. This result shows that other causal mutations are not likely to exist in the tested individuals.

We tested the possibility that natural selection on the ABCC2 gene is responsible for the field-evolved Bt resistance in the Brazilian population. In this case, ABCC2 is expected to have increased genetic differentiation between resistant and susceptible groups. F_ST_ calculated between these two groups was 0.2922 at ABCC2 gene. In total, 7.7% of 100,000 randomly chosen loci with the same length of the ABCC2 gene had F_ST_ higher than 0.2922 (equivalent to *p*-value = 0.077) (Figure 3), indicating marginally significant statistical support for increased genetic differentiation. This result implies that natural selection might have increased the Bt resistance of the Brazilian population.

### 3.2. Synthetic Insecticide Resistance

P450 genes increase the level of synthetic insecticide resistance in FAW, and P450 gene duplication has been suggested as a possible cause underpinning resistance [23]. In total, 99 P450 genes were identified from the reference genome (Appendix A). Copy number variation compared with the reference genome was observed from 47 P450 genes, which include4, 12, and 22 genes belonging to clan2, clan3, and clan4, respectively. In all clades, invasive populations have higher numbers of duplicated P450 genes per individual than native populations for all clades (Figure 4). On the other hand, native populations have higher numbers of deleted P450 genes per individuals than invasive populations for all clades. Consequently, invasive populations have overrepresented numbers of duplicated P450 genes compared with deleted ones than the native populations (*p*-value = 6.018 × 10^−21^, 0.0004553, 0.04414 for clan2, clan3, and clan4, respectively; two-tailed Fisher’s exact test). This result supports the hypothesis that invasive populations have higher P450 gene numbers than native populations through CNVs.

The resequencing data show all three reported resistance mutations at the AChE (Figure 5), as shown by Boaventura et al. [34]. F209V mutations were observed from populations in Uganda, Malawi, Brazil, and China, as reported from Guan et al. [21] as well as India, Benin, Puerto Rico, Mexico, Mississippi, and Florida. Invasive populations have a higher proportion of individuals with resistance F209V mutations (80.26%) than native populations (72.37%) (*p* = 0.001097, two-tailed Fisher’s exact test). A201S mutation was found from populations in Uganda, Malawi, and Brazil, as shown by Guan et al., but also from Benin, Mississippi, and Florida. Invasive populations have a higher proportion of individuals with A201S resistance mutations (20.0%) than native populations (7.14%) (*p* = 0.01966). Interestingly, two codons were found for the susceptible ‘A’ in the population from Benin (GCG and GCA). In Benin, the GCG codon was observed from 28 individuals (as well as other populations). The GCA codon was observed from three individuals in Benin. The most parsimonious explanation is that the GCA codon was newly generated from GCG by a point mutation at the third codon position. The G227A mutation was found in Brazil, as per Guan et al. [21] as well as in Puerto Rico and Florida. The proportion of individuals with this resistance mutation was 17.65% in native populations. Resistance mutations at RyR (I4734M and G4946E substitutions) or the VGSC (T929I, L932F, and L1014F substitutions) were not observed in the analyzed populations of FAW.

## 4. Discussion

The global spread of FAW through invasion may be accompanied by the spread of resistance mutations to Bt or synthetic insecticides to invasive areas. This is known to occur in other species. The population of the cotton bollworm *Helicoverpa armigera* is thought to have invaded South America with a pyrethroid resistance allele, which was then shared with a sister species, *H. zea* [43,44]. Therefore, monitoring the geographical distribution of resistance mutations in the context of invasion can provide useful information for pest FAW control and insecticide resistance management. We identified known resistance mutations from the whole genome sequences of 177 individuals collected from 12 geographic populations in North America, South America, Africa, India, and China. We showed that resistance Bt mutations at ABCC2 remained at their origin in Puerto Rico and Brazil, meaning that these mutations do not spread to invasive areas. However, invasive populations have increased copy numbers of P450 genes compared with native populations through copy number variations. We also observed that invasive populations had increased proportions of resistance mutations at AChE. These results can be best explained by selective pressure causing increased resistance against synthetic insecticides in invasive populations.

Bioassay experiments demonstrated that Chinese populations are susceptible to Bt Cry1Fa toxins [31]. The analyzed Chinese samples in this study did not have mutations causing Cry1Fa resistance, explaining the susceptibility to this Bt toxin. Interestingly, a South African population showed moderate resistance against Bt Cry1A [32], while GC insertion causing Cry1A resistance [18,19] was not identified from invasive populations in the resequencing data. Therefore, the South African population could potentially have a resistance mutation that is yet to be characterized in native populations, and we cannot exclude the possibility of on-going gene flow of unidentified resistance mutations either from native to invasive populations or from other invasive populations that represent independent introduction from South African FAWs. More studies are necessary to identify causal resistance mutation in the South African populations and to identify the origin of this resistance mutation.

If a resistance Bt mutation exists in invasive populations with low frequency, individuals with the resistance mutations might not be included in the analyzed resequencing data. If this is true, existing resistance mutations in invasive populations might be undetected. The probability of not finding a resistance mutation is (*1 − p*)*^2N^*, where *p* is the allele frequency of resistance mutations and *N* is the number of diploid organisms. If the allele frequency of resistance Bt mutation is 0.01, representing a rare allele, the probability of missing this mutation in our resequencing data is only (1 − 0.01)*^2 × 99^* = 13.67%. Therefore, we do not believe that a resistance Bt mutation is likely to exist. Even in the case that this possibility of 13.67% is true, the resistance mutation is likely to be eliminated in a population by genetic drift due to the low allele frequency (0.01).

A whole-genome analysis is necessary to contrast the abundance of P450 genes between invasive and native populations, because P450 genes exist in multi-copies scattered across the genome [9,10,11]. Our whole genome analysis at a population level shows an increased copy number of P450 genes in the invasive FAW populations. Bioassay experiments demonstrated that Chinese populations have resistance to pyrethroid insecticides [32,45]. As P450 genes are known to cause resistance against deltamethrin [23], a type of pyrethroid in FAWs, the increased P450 gene numbers might be responsible for this resistance. Chinese populations also have resistance against organophosphate [32,45] as well as oxadiazine and diamide [32]. As P450 proteins are probably able to detoxify other types of insecticides than pyrethroid, the increased P450 genes may be responsible for the resistance against a wider range of insecticides.

In addition, the increased proportion of resistance mutations at AChE can cause the increased resistance against synthetic insecticides in invasive populations. Multiple resistance mutations with the increased proportion imply strong selective pressure for resistance against synthetic insecticide. The vast majority of beneficial mutations are quickly removed in a population due to stochastic fluctuation of allele frequency by the genetic bottleneck, and the survival probability of beneficial mutations is just two times that of the selective coefficient. Therefore, the probability of observing a single resistance mutation at AChE is proportional to the extent of increased fitness due to the resistance. The probability of observing multiple resistance mutations is a function of the multiplication of each selective coefficient, ranging between 0 to 1, at each resistance mutation. Therefore, multiple resistance mutations are difficult to observe without very high selective coefficients, and we argue that invasive populations are under strong selective pressure for the resistance against synthetic insecticides.

Glutathione S transferases or UDP glycosyltransferase might also cause resistance against synthetic insecticides. However, the direct relationship between these two gene families and resistance is yet to be tested in FAW. Genomic analyses and bioassay experiments will generate a more comprehensive list of causal resistance mutations against synthetic insecticides.

The differential usage of synthetic insecticides may explain the observations that invasive populations have increased allele frequency of resistance mutations at AchE genes and increased number of P450 genes by CNVs. The average annual insecticide use is particularly high in East Asia including China [46]. We speculate that initially introduced FAW insects in East Asia might have survived through adaptive evolution by the increased P450 gene number or the increased allele frequency of resistance mutations at AchE genes, while these FAWs remained undetected due to the low population sizes before explosive population expansion in West Africa, where genetic admixture occurred between populations with different invasive origins [31]. More studies are necessary to test the role of adaptive evolution associated with synthetic insecticide resistance in invasive success.

The F_ST_ result supports natural selection for Bt resistance in the Brazilian population. If selective pressure with the same directionality exists in invasive populations, newly generated Bt resistance mutations can be selected in the near future. Theoretically, species with large population sizes have higher adaptive potential than species with small population sizes because of a higher population-scaled rate of mutation [47,48]. Since the estimated population size of FAWs is as high as 3.75 million [49], the population-scaled mutation rate can be sufficient to generate a genetic variation for drastically rapid Darwinian positive selection. We observed two synonymous codons at a resistance position in AChE from a population Benin, implying that mutational influx could not be a limiting factor of adaptive evolution through positive selection. Therefore, invasive populations might experience adaptive evolution, causing resistance against Bt mutations in the near future.

## 5. Conclusions

In this study, we showed that Bt resistance mutations did not spread during this period to the Old World by the surveyed invasive FAW populations and that invasive populations have increased copy numbers of P450 genes and increased proportions of resistance mutations at AchE compared with native populations. These results imply that invasive populations might be under selective pressure for the resistance against synthetic insecticides.

The samples used in this study were obtained between 2017 to 2018. Since the first reported FAWs in Western Africa in 2016 [31], the genetic information in this study represents snapshots at the early phase of invasions. As Bt crops and synthetic insecticides are commonly used in the invasive area, the level of insecticide resistance can be changed together with underlying genotypes. This change is likely because our result is in line with selective pressure both for Bt and, in particular, synthetic insecticides. Therefore, monitoring changes in field susceptibility to various insecticides will be helpful in FAW control, together with the analysis of changes in genomics sequences to identify known or new causal resistance mutations.

## Figures and Tables

**Figure 1 insects-12-00468-f001:**
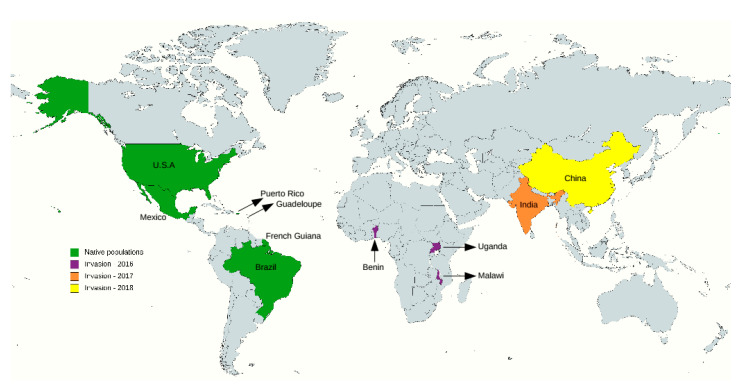
Countries where the analyzed individuals were sampled. The green color indicates native populations. Purple (2016), orange (2017), and yellow colors (2018) indicate the reported years of detection in countries from where individuals were collected. The map was generated using mapchart [37].

**Figure 2 insects-12-00468-f002:**
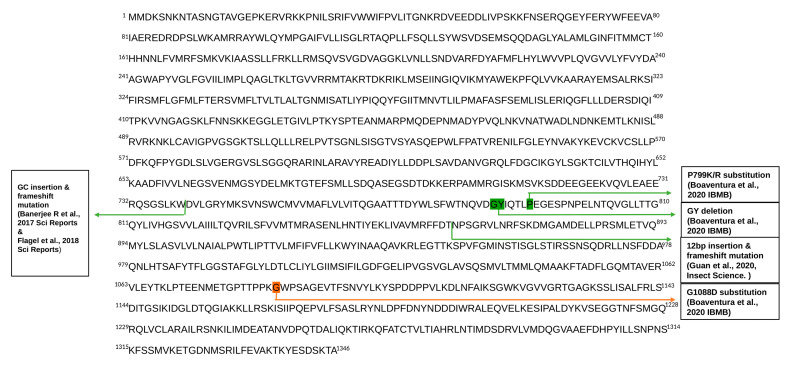
Full amino acid sequence of the ABCC2 gene with information on all mutations studied. The green and orange arrows indicate the identified and unidentified mutations from the resequencing data, respectively.

**Figure 3 insects-12-00468-f003:**
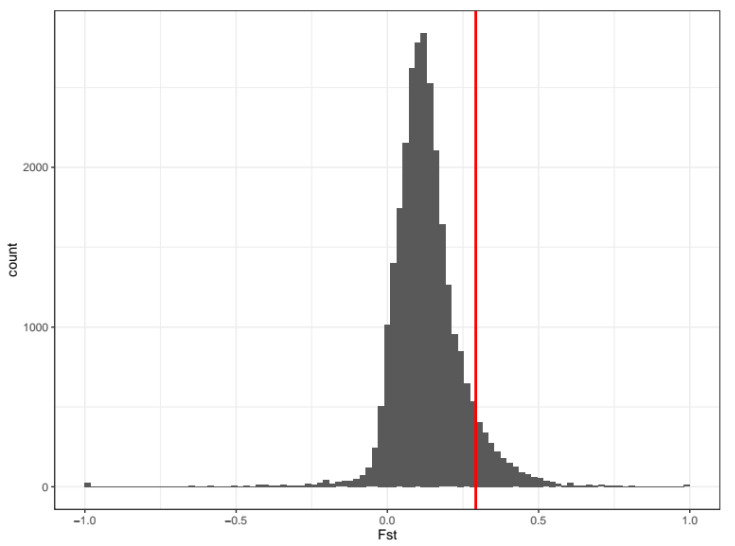
F_ST_ calculated between CC and rCC (red vertical line) and from random grouping among CC and rCC.

**Figure 4 insects-12-00468-f004:**
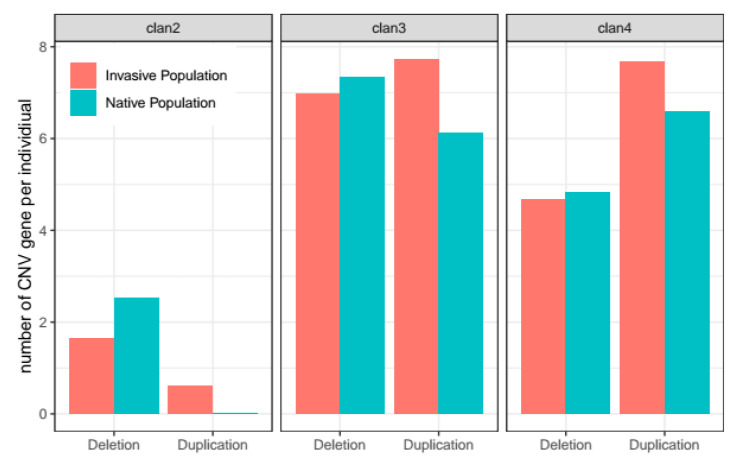
Average numbers of deleted or duplicated P450 genes per individual in invasive and native populations for clan2, clan3, and clan4.

**Figure 5 insects-12-00468-f005:**
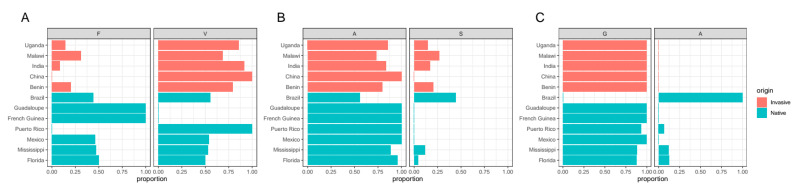
The proportion of individuals with susceptibility and resistance at AChE for (**A**) F209V, (**B**) A201S, and (**C**) G227A in invasive and native populations.

**Table 1 insects-12-00468-t001:** Individuals with observed resistant mutations (* indicates individuals heterozygous for the mutations). PR: Puerto Rico, CC: susceptible Brazilian individuals, rCC: resistant Brazilian individuals.

Mutations	Individuals with Resistance Mutations
2 bp insertion and a frameshift mutation	* PR1, PR5, PR12, PR14, PR16, * PR18, * PR19, * PR27, PR30, * PR31, * PR33
GY deletion	CC44, CC69
P799K/R	CC44, CC69
12 bp insertion and a frameshift mutation	* rCC25, * rCC5

## Data Availability

We will provide all files generated in this study upon request. These files include vcf files, gene annotation files, and computer programming scripts.

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
