# Peer review of "Geographic Monitoring of Insecticide Resistance Mutations in Native and Invasive Populations of the Fall Armyworm"

_insects, 2021, doi:10.3390/insects12050468_

Round 1

Reviewer 1 Report

Geographic monitoring of insecticide resistance mutations in 2 native and invasive populations of the fall armyworm In this study authors describe status of resistance to Bt toxin and synthetic insecticides in invasive populations of fall armyworm through analysis of previously published genomic data for 177 individuals collected from 12 geographical locations. The authors find that the Bt resistance mechanisms did not appear to have crossed during the invasion but the mutations in AchE gene causing resistance to organophosphates are present in tested individuals. Overall the conclusions of the manuscript are sound, however the paper does feel a bit shallow. I think the data itself is useful to warrant a publication but certain sections should be expanded to increase the overall value of the manuscript. Please see below of some of my main concerns: What is the general profile of used insecticides between different locations? The reviewer assumes the control methods are different between Africa and other regions due to economic costs. This information might affect the potential path of insecticide resistance development in tested regions. Are there many silent SNPs identified between tested individuals in the genes of interest AchE, VGSC and RyR. FAW is known to be fairly polymorphic and a snap shot summary of the degrees of variability between various known target site genes would be useful. P450 genes are indeed a diverse group of enzymes. Simply stating that a given number of them is modified is not particularly informative although it might imply adaptation to new environments/insecticide regimes. Authors should expand on this section and at least specify to which of the 4 main classes of CYPs the gene duplications refer to. Since the authors have a full list of all identified P450s a figure which includes the phylogenetic tree highlighting potentially duplicated genes would be incredibly useful and should be included in the manuscript. If authors have access to any RNAseq data for various populations of this species any inclusion of potential shifts of expression would add a lot of value into the paper. Minor: Figure 4 appears not be referenced in the text and the differences in the respective columns appear minimal. The conclusions the authors want to read from this figure are not clear/or really supported. Please either delete or amend the figure so that the perceived statistically significant change can be more clearly visible. Colour scheme/legend for the figure 5 should be redesigned. Especially the invasive/native distinction lacks clarity. Line 78-79 correct to five mutations causing Bt resistance.

Reviewer 2 Report

It is a very good manuscript which provide novel facts. The manuscript is well written and organized, the molecular and statistic analyses are adequate.

I have not any major comment, and my recommendation is that the manuscript should be accepted for publication.

Below there are a few minor comments that the author should take into consideration.

-Apart from insecticide resistance mutations, you have examined copy number variation in P450 genes, maybe the latter should be also highlighted in the title of the manuscript.
-Introduction, line 58. Why you mention only these old chemical classes of insecticides?
-Introduction, line 68. In insect pests, there are resistance mutations in various target proteins, not only in calcium channels. You could provide additional examples.
-Introduction, lines 91-94: mutations in VCGS confer resistance to pyrethroids. In this sentence you report only organophosphate and carbamates. 

Reviewer 3 Report

The fall armyworm is a serious agricultural pest and has developed resistance against Bt toxins as well as synthetic insecticides. The fall armyworm is native to America, but has invaded Africa, Asia and Australia since 2016. Authors here examined the geographical distribution of mutations causing BT and insecticide resistance using 177 individuals from 12 geographic populations collected for an earlier study and reuse of their sequencing data. Their results demonstrate that BT resistance mutations (at ABCC2 genes) are found in native populations, where invasive populations have a higher copy number of Cyt P 450 genes and more resistance mutations at AChE genes, which means higher resistance to synthetic insecticides. 

The results are of high practical importance for fall armyworm control and should therefore be published, although the present manuscript is not result of experimental work in a narrow sense..

The manuscript is generally well written but needs some editorial improvement:

Authors should say when and where the samples were exactly collected for the previous study. I miss Conclusions at the end of Discussion. References have to be presented according to MDPI author's instructions.

  • line 17: do not mention a specific country here, but regions as in line 31
  • line 56: give species names in italics
  • line 67: resistance against synthetic insecticides...
  • line 72: 76 plant families...
  • line 78: authors probably mean Bt resistance
  • line 88: insecticides
  • line 219: the resequencing data show.... ?
  • line 225 and others: as shown by Guan et al.
